# Sustainable Technology Strategies for Transportation and Logistics Challenges: An Implementation Feasibility Study

**Muhammad Saleem Sumbal [1], Waqas Ahmed [2], Huzeifa Shahzeb [2] and Felix Chan [3,*]**

1   Department of Industrial and Systems Engineering, Hong Kong Polytechnic University, Hong Kong, China; saleemkhan.sumbal@polyu.edu.hk

2   Operations and Supply Chain Department, National University of Sciences and Technology, Islamabad 44000, Pakistan; engrwaqas284@gmail.com (W.A.); huzaefa_zeb@nustedupk0.onmicrosoft.com (H.S.)

3   School of Business, Macau University of Science and Technology, Macau 999078, China

*   Correspondence: tschan@must.edu.mo

**Abstract:** Transportation and logistics are the basic building blocks in the socio-economic development of a country. The pandemic altered the landscape of the transportation and logistics sector where organizations had to look for new technology-based solutions. Block chain and digital trucking are emerging concepts, which were further accelerated by COVID-19, to manage the challenges in the transportation and supply chain industry. This study, therefore, investigates the challenges faced by the transportation industry during and post COVID-19 and, consequently, identifies relevant sustainable strategies to combat these challenges in a developing-economies context. Data were collected through interviews from 20 key personnel working in managerial positions in the transportation industry of Pakistan, a developing economy. The findings of this study indicate that the challenges faced by the transportation industry are reduced import–export, local market orders and revenues, supply limited to a few edible items, increase in e-commerce, new entrants in the market and operational issues, such as lack of standard operating procedures (SOPs), performance management and training of drivers. These challenges were more pronounced in the COVID-19 era; however, they are still impacting the industry. Thus, in the post-COVID-19 era, transportation companies need to opt for efficient strategies, such as contactless deliveries, expansion in e-commerce, tech-based performance management of drivers and digital trucking for sustainability, in a developing economy's transportation and logistics sector.

**Keywords:** logistics; transportation; COVID-19; sustainable strategies; challenges; Pakistan

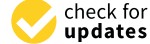



## 1. Introduction

Transportation and logistics are the most important factors worldwide for economic development and for a country's success [1,2]. In modern times, where technology is at the forefront in all fields of life, the transportation and logistics sector is also embracing technological solutions for enhanced performance [3]. This technology adoption was further fueled by the COVID-19 pandemic [4], specifically for developing economies [5]. COVID-19 brought additional challenges and added complexities in the transportation and logistics sector in developing economies. On one hand, there were concerns related to technological aspects, such as e-commerce and smart supply chain management [6], and, on the other hand, there were challenges faced by human resources (specifically crew involved) in the transportation industry of developing economies [7–9]. In the context of developing economies, several factors, such as a lack of investment and proficient policies, corruption, lack of technology adoption, inadequate resources and infrastructure, etc. [7,10–12], affect operations in the transportation industry. In addition, political reasons, natural disasters and market fluctuations [4] also have a profound impact on this sector. COVID-19 was one such disaster, which had a devastating impact on developing economies [13] with already weak and

underdeveloped transportation and logistics infrastructure, as compared to developed economies. An important factor of concern is that COVID-19 impacted all places differently due to factors, such as working environments, resource availability, infrastructure and severity of the pandemic [14]. Hence, the role of context is very important to understand the challenges and strategies in certain perspectives such as a developing economy.

Pakistan is a developing economy, and the logistics industry is one of its most thriving industries [1]. It has made some significant progress to develop into a profitable and well-structured industry in the last decade through infrastructure developments, such as roads, highways and better warehousing facilities [10]. In this context, a significant role has been played by the China Pakistan Economic Corridor (CPEC) under China's Belt and Road initiative [15]. The strategic location of the country plays a critical role in this Belt and Road initiative, and its success is linked with the development of roads, railways and highway networks, oil and gas pipelines and the transportation industry [16]. The existing literature focuses on the supply chain and logistics of the country but to a limited extent [1,15,17–19] and largely ignores the COVID-19 and post-COVID-19 context in relation to the transportation sector, which was severely impacted. Apart from the strategic importance of CPEC in relation to transport, the supply chain and the logistics industry, certain dimensions of the supply chain and logistics in Pakistan demand exclusive and in-depth exploration of the linked phenomenon, i.e., illiteracy of the personnel working in the transportation industry and the local truck management system at stations called "Addas" without much focus on technology. Because of the limited resources and inadequate economic environment [20], the aforementioned areas require special focus. The development of the China Pakistan Economic Corridor (CPEC) has added extra value and pace to this industry. Contractual and permanent, full time and part time, all kinds of employment are seen in the transportation sector of Pakistan. Therefore, the parallel management of a diverse workforce is a challenge in the transportation industry. Although human resources and operations in the transportation industry have improved in the last few years, the effective use of technology is lacking, as well as well-planned human resource processes and operation execution [12]. The COVID-19 pandemic and post-pandemic scenarios brought additional challenges due to the shutdown of businesses. This study, therefore, investigates the human resource practices in Pakistan's logistics industry, especially during and in the post-COVID-19 context. This study contributes to the literature on workforce challenges and the adoption of technology in the logistics sector. It highlights the significance of technology literacy among the conventional setup of the transport sector in a developing economy. Although initiatives such as the Belt and Road initiative are drivers for the economy, the technology illiteracy of owners and employees affects operations in the transport sector. Moreover, it is hard to grasp the opportunities, specifically for smaller businesses, under such conditions. New entrants can grasp these opportunities and market share due to a better understating and technology literacy. Through the identification of workforce challenges, this study further contributes towards sustainable strategies for the industry, which could help managers and executives to efficiently manage their workforce and add value to transportation businesses in Pakistan. The rest of this article is organized as follows. Section 2 covers the literature review. Section 3 discusses the methodology. Section 4 elaborates on the findings of the study, followed by the conclusion in Section 5.

## 2. Literature Review

### 2.1. COVID-19 and Logistics Industry

The COVID-19 pandemic brought incredible difficulties to the logistics industry [21–23], including a sharp drop in logistics demand, disturbance of the logistics network and expansion in operational expenses and the number of loss-making ventures [10]. Post-pandemic, the logistics industry has grown quickly, prompting critical changes, such as demand for logistics, supply of logistics, logistics foundation or infrastructure, logistics information and logistics industry improvement and development [24]. The pandemic brought in the concept of contactless deliveries, which refers to a delivery process in

which no physical contact exists between the driver, the customer and store or warehouse employees [25,26]. On one hand, contactless delivery focuses on the safety of staff, their wellbeing and the use of extra storage spaces, whereas, on the other hand, it requires more contactless offices; thus, the underlying fixed venture cost of the activity is higher than the previous methods, for example, the sanitization cost of delivery bundles and the development cost of contactless conveyance offices. As per data published by the CFLP (China federation of logistics and purchasing) on 20 May 2020, many logistics organizations suffered great losses due to these additional costs. Particularly, 57.6 percent of Small-Medium Enterprises were affected with losses [27]. Thus, the concept of contactless deliveries was of high importance during the pandemic and remains so today [25]. With respect to the global market, numerous nations shut their borders and international logistics was enormously limited. As the pandemic was progressively taken care of, worldwide transportation started again normally [5]. Smart logistics technology is a new driving force in the logistics industry, fueled by the COVID-19 outbreak. Smart logistics refers to a system, which involves the application of smart technologies and devices for decision making, analysis and the execution of logistic operations [28,29].

### 2.2. Challenges and Driving Forces in the Logistics Industry

Several supply chain challenges have been identified in the literature in relation to COVID-19 and post-pandemic circumstances. During COVID-19, organizations faced logistics issues, such as pressure from buyers on reducing delivery time, real-time forecasting complexities, bankruptcy of various partners, shortage of physical and financial resources, decrease in demand, layoffs of highly paid workers, employees' demand for sustainability, changes in distribution networks, shortage of skilled manpower, slower productivity and payment withholding from buyers [30]. Covering various economic, social, technological, political and organizational aspects of the logistics sector, Eryarsoy et al. [31] identified challenges, such as trust issues, opportunism, dependency, competition, lack of knowledge, complex network coordination, complex system implementation, strategic management, financial and human resources, employees' concerns, etc. During the pandemic era, the focus of the supply chain also shifted towards COVID-19 vaccines and related materials. The key challenges in this context were vaccination cost and lack of financial support, limited number of vaccine manufacturing companies, lack of accurate forecasting for vaccine demand, behavioral challenges, i.e., consumer unwillingness to vaccinate, inadequate positive vaccine marketing, unavailability of people for vaccine trials, lack of planning and scheduling, increase in acquisition lead time, cold chain storage challenges, vaccine temperature monitoring and control, organizational challenges, such as inadequate coordination, managing monitoring bodies and lack of correspondence between virtual supply chain members [32]. Pujawan et al. [33], in the same direction, identified the challenges in achieving optimized distribution, prioritization of social groups, manufacturing low-cost vaccines, temperature control in remote areas and less-developed regions, etc. In a developing-economy perspective, the highlighted supply chain challenges during the pandemic along with coping strategies are uncertainty of demand, inconsistent supply, scarcity of material, delivery delays, suboptimal substitute adoption, labor scarcity, suboptimal manufacturing, storage capacity constraints, delays, vehicle unavailability and last-mile delivery challenges [34]. The authors also identified long- and short-term strategies to overcome these challenges in terms of capabilities and supply and demand in supply chains and logistics.

The supply chain and logistics are regarded as a new stage of globalization, along with the concepts of regionalization or continentalization [2]. In this direction, there are important factors that lead to modern advancements and improvements in the transportation industry, especially logistics and the supply chain. With respect to China's logistics industry, three main factors have been noticed, i.e., request pull, technology push and strategy support [3,35,36]. The role of technology is pivotal in this context. Smart logistics incorporates technology, such as Unmanned Aerial Vehicles (UAVs), savvy package

storage spaces, cloud computing, block chain and AI, to execute operations and enhanced performance in the logistics industry [37,38]. Bigger economies, such as China, experienced changed patterns in five dimensions, i.e., logistics request, logistics supply, logistics framework, logistics information and logistics industry improvement [39]. Similar impacts can be expected in other parts of the world, including developing economies. The advent of new business models during COVID-19 broadened the logistics horizon. The concept of contactless deliveries will speed up the development of an ever-increasing number of contactless offices, for example, shrewd package storage spaces and conveyance robots/UAVs [40]. Recent studies indicate the need to develop frameworks to guarantee the adaptability of technology in order to speed up the processes and productivity in businesses [41–43]. In the case of developing economies, this study adds to the literature on workforce challenges and the adoption of technology in the logistics sector. The adoption of technology is important for the workforce to remain competitive and carry out operations in a better way. Weak technological infrastructure and unfamiliarity as well as a lack of training for the workforce can further enhance these barriers. Thus, this study aims to cover this perspective.

## 3. Methodology

The focus of this study is the transportation industry; however, it is limited to road freight logistics and inbound and outbound logistics, details of which are provided in Table 1. To explore the challenges faced by transport industry during and after COVID-19 and for the identification of relevant sustainable strategies, a qualitative research design was adopted as the aim was to gather rich insights related to the transportation sector through the experience of people involved in this industry and in the pertinent context of a developing economy [44]. Primary data were gathered through face-to-face and online interviews using an interview guide.

**Table 1.** Explanation of freight logistics and inbound and outbound logistics.

| | |
|---|---|
| Inbound Logistics | Logistics that carries raw material from market to production plant. |
| Out Bound Logistics | Logistics that carries finished goods from production plant to other factories and warehouses. |
| Primary Logistics | Logistics from factory to warehouse is known as primary logistics. |
| Secondary Logistics | Logistics from warehouse to supermarket is known as secondary logistics. |

*Data Collection and Analysis*

Qualitative data collection, such as interviews, focus groups, etc., is viewed as appropriate for a deeper understanding of the phenomenon under study. Interviews are an efficient qualitative data collection method when point-to-point experiences are expected from individual members [45]. Semi-structured interviews were conducted with managers, CEOs and owners of different transport companies. An interview guide with open-ended questions was developed so that respondents could provide detailed insights and share their experiences. The interview guide was developed based on the study objective, existing research work and through consultation with two experts from transport industry. As the interviews progressed, some of the questions were modified based on the feedback and information shared by respondents. Respondents were chosen based on the contact points of one of the authors who works in the logistics sector. Further, snowball sampling was used to select key respondents for the study. The target participants were contacted through WhatsApp, emails and phone calls. Based on availability, a total of 20 interviews were conducted, in line with other qualitative studies in similar contexts [46]. On average, each interview lasted between 50 and 70 min, depending on the responses received from respondents. Prior to the interview, the interview guide was shared with the respondents to have a look at the questions and if they had any queries regarding the questions. The interviews were recorded and interpreted subsequently. Generally, participants were directly or indirectly related to the logistics industry. The details of those directly and indirectly related

to logistics industry are given in Tables 2 and 3, respectively. This study is intended to cover the challenges faced by people directly related to logistics businesses. Organizations included in the study are from pharmaceutical industry, fast-moving consumer goods (FMCGs), food and beverage industry, oil companies and logistics companies from ten major cities around the country. These organizations were selected based on the relevance and suitability in relation to the research objective of the study.

**Table 2.** People directly related to logistics business.

| People Directly Related to Logistics Business. | Responsibilities |
| --- | --- |
| Driver | Two drivers on a truck are responsible for vehicle safe transit. |
| Helper | To help driver complete the journey within time. |
| Supervisor | Supervising a small group of trucks normally a fleet of 10 trucks. |
| Workshop Supervisor | Caretaker of truck repair and maintenance. |
| Assistant Manager | Responsible for all smooth operations of logistics including closing of trip. |
| Logistics Manager | Heading a region with a small fleet size providing 3PL service to customer. |
| Regional Manager | Heading and controlling several regions. |

**Table 3.** People indirectly related to logistics business.

| People Indirectly Related to Logistics Business. | Descriptions |
| --- | --- |
| Local Station "Addas" | They are brokers that arrange supply orders for transporters to lift and deliver stock to a customer's premises. They personally don't have any fleet of trucks. |
| Computerized Weight Balance | They provide facility to check and confirm weight of loaded and empty vehicle. |
| Labor | Group of people that load and unload trucks in warehouses or factory premises. |
| Fast Moving Consumer Goods | Includes all companies that produce products for end consumer. |
| Exercise & Taxation | Authority that holds power to permit or stop any truck due to lack of documentation of route permit, fitness certificate, insurance certificates and vehicle token tax. |
| Traffic Police | Responsible for penalizing any truck driver due to violation of traffic rules. |

Table 4 presents the profile of the respondents from these organizations. It is evident that respondents are the "elite informant" and have wealth of experience in order to answer the interview questions. Furthermore, all the respondents had more than 7 years of experience, and majority of them were in managerial positions. Data were analyzed by transcribing the interview and performing content analysis of the data. Line-by-line analysis was performed to generate the relevant codes from the data. The coding was performed using two stages: initial coding and focused coding. Initial coding comprised coding

all the data from the transcribed interview, whereas focused coding involved choosing codes relevant to the study objectives. The themes were generated in two dimensions, i.e., challenges faced by the logistics and supply chain industry in Pakistan and strategies for sustainable performance and growth in the industry. In addition to recording and transcription, notes were also taken during the interviews and later matched with the findings of the interview as well as being used for integrating all the emerging themes for analysis purposes. Furthermore, the findings of this study were also compared with the existing literature. These measures helped in enhancing the reliability and validity of the findings [47].

**Table 4.** Respondents' profile.

| Interviewee | City | Organization | Designation | Experience |
|---|---|---|---|---|
| F1 | Faisalabad | B | Head Logistics | 15 |
| H1 | Hattar | O | Operations Manager | 10 |
| I1 | Islamabad | A | Manager Logistics | 12 |
| I2 | Islamabad | C | Area Sales Manager | 10 |
| I3 | Islamabad | G | Branch Manager | 11 |
| I4 | Islamabad | K | Operations Manager | 18 |
| I5 | Islamabad | L | Supply Chain Manager | 19 |
| I6 | Islamabad | M | Head of Logistics | 13 |
| K1 | Karachi | T | Logistics Manager | 12 |
| L1 | Lahore | E | Supply Chain Manager | 15 |
| L2 | Lahore | F | Supervisor | 7 |
| L3 | Lahore | R | Manager Operations | 9 |
| R1 | Rahim Yar Khan | D | Logistics Manager | 16 |
| R2 | Rawalpindi | S | Supervisor | 15 |
| S1 | Sheikhupura | J | Manager Operations | 9 |
| S2 | Sheikhupura | H | Area Sales Manager | 15 |
| S3 | Sheikhupura | N | Operations Manager | 13 |
| S4 | Sheikhupura | P | Physical Logistics Manager | 17 |
| S5 | Sialkot | Q | Supervisor | 12 |
| T1 | Tarnol | I | Operations Manager | 10 |

## 4. Findings and Discussion

This section will detail the findings of the study, mainly related to the challenges and strategies of the companies during and post COVID-19.

### 4.1. Challenges Faced by Logistics Sector

#### 4.1.1. Declining Revenues

Logistics companies faced great losses during COVID-19 because of reduced business activities. According to the respondents, due to the Government of Pakistan's lockdown policies during the pandemic, all businesses were either temporarily closed or limited their business processes, which affected their revenue. All types of dispatches, except food and beverages (F&B), business to business (B2B), business to customer (B2C), customer to business (C2B) and primary and secondary dispatches, were stopped. Strict bans and strategies implemented all over cities ultimately affected logistics operations [35]. This resulted in low customer transportation orders to third party logistics providers as well as lower margins and declining profits, especially in transportation and logistics. H1 described this as follows:

> "*So, if we see this scenario, all type of dispatches of logistics including primary, secondary, inbound, outbound, all dispatches were declined because of production units and warehouses were closed which effected the businesses.*"

#### 4.1.2. Reduced Import and Export

FMCGs working in Pakistan purchased their raw material from other countries, mostly from China. During COVID-19, a lot of countries including China banned movement across borders to control the virus spread. This ultimately reduced import and export orders in every country including Pakistan, which resulted in zero or reduced dispatched orders from port Qasim (main port in Karachi) to manufacturing plants and vice versa. L1 explained this as follows:

> "*Second import, export orders reduced because of worldwide bans at all geographical locations . . . as initially the virus started from China . . . within Pakistan closure of Port Qasim as well as clearance issues related to customs impacted the (logistics) industry.*"

#### 4.1.3. Reduced Local Market Dispatch Orders across Pakistan

The first lockdown was implemented by the Government of Sindh in the province for 14 days from 23 March 2019, ordering all public transport, markets and offices to stop their operational activities. Later, smart lockdowns were imposed in all provinces and cities, depending on virus-affected localities. Moreover, the public in Pakistan started to store products and items used for daily needs because of the fear that markets would be closed for the lockdown period and there would be a restricted supply of consumer products, and these products could have become expensive due to a shortage of food and grocery items. Respondents revealed that virus effects, fear of virus and lockdown resulted in labor shortages, which ultimately impacted local markets and transportation. Thus, there was an inconsistency in the supply chain, which is also evident even after the pandemic [34].

#### 4.1.4. Limited Supplies—Food and Beverages Only

As lockdown continued, the city and provincial government immediately felt a need to make a policy regarding food and beverage supplies across Pakistan. The government advised all FMCGs to start food processing units, which had previously been stopped during lockdowns. The shortage of raw materials and the labor shortage also created a big bottleneck. Every food processing unit made a special COVID-19 policy to run production units and resumed food supplies operations from factories to markets through improved supply chains and transportation networks. Different types of supplies have an impact on logistics differently, especially during pandemics; therefore, it is important to understand the types of supplies in order to decide on the logistics and supply chain. The difference

can be seen in Table 5. Hence, according to the respondents, some supplies increased drastically (e.g., medicine and health care), whereas some reduced, such as raw materials and local market orders of other products, which are not essential for living, for example, construction materials. Respondent R2 explained this as follows:

> "*During COVID-19, if we see that only the industries that were generating the logistic dispatch orders were affinity food and beverage industries, that was also globally running whereas import and export orders were reduced, local market orders were limited, raw material orders were reduced. Hygiene product orders were increased across Pakistan.*"

**Table 5.** Impact of supplies during COVID-19.

| Type of Supplies | Logistic Operations during COVID-19 |
| --- | --- |
| Food and Beverage | Only supplies which were running globally |
| Import and Export | Reduced supplies |
| Local Market Orders | Limited orders |
| Raw material | Reduced and limited |
| Medicine and devices | Increased orders |
| Health and Care | Increased orders |
| E-Commerce | Increased orders |

### 4.1.5. Increased E-Commerce Business

COVID-19 forced businesses and the public to move quickly towards e-commerce for trade and purchase. Organizations shifted towards e-commerce platforms in Pakistan, and e-commerce businesses exponentially increased in just a few months during the pandemic [13]. Some of the common e-commerce companies working in Pakistan are listed in Table 6. Moreover, the buyer decision-making process changed dramatically during and after the pandemic. It is now more oriented to online shopping and the delivery of basic items, such as groceries, food, medicines, fruits and vegetables, etc.

**Table 6.** Names of e-commerce companies.

| Sr. No | E-Commerce Companies | Websites |
| --- | --- | --- |
| 1 | Daraz | Daraz.pk |
| 2 | Dastgyr | dastgyr.com |
| 3 | Tajir | tajir.app |
| 4 | Airlift | airlifttech.com |
| 5 | Bazaar | bazaar-tech.com |
| 6 | Food Panda | foodpanda.com |
| 7 | Cheetay | cheetay.pk |
| 8 | Jovi | jovi-app.com |
| 9 | Jugnu | jugnu.pk |
| 10 | Goto | Goto.com.pk |
| 11 | Ali Express | Aliexpress.com |
| 12 | Telemart | Telemart.pk |
| 13 | Shopon | Shopon.pk |

**Table 6.** *Cont.*

| | Sr. No | E-Commerce Companies | Websites |
|---|---|---|---|
| 14 | | Homeshopping | Homeshopping.pk |
| 15 | | Ishopping | ishopping.pk |
| 16 | | Yayvo | Yayvo.com |
| 17 | | Symbios | Symbios.pk |
| 18 | | Vmart | Vmart.pk |
| 19 | | iBucket | iBucket.pk |
| 20 | | Krave Mart | Kravemart.com |
| 21 | | Amazon | amazon.com |
| 22 | | Dawaai | Dawaai.pk |
| 23 | | Grocer App | GrocerApp.pk |
| 24 | | Price oye | Priceoye.pk |
| 25 | | Baby Plannet | BabyPlannet.pk |
| 26 | | 24seven | 24seven.pk |
| 27 | | Shopistan | Shopistan.pk |
| 28 | | Kamyu | kamyu.pk |
| 29 | | Medical Store | MedicalStore.pk |
| 30 | | Lootlo | lootlo.pk |

Buyers conduct extensive research online before speaking to a salesperson or visiting a shop. The internet makes business easier and faster as buyers make more direct purchases online via smart phones and do not physically visit shops and stores [35]. Online stores decreased the need for warehouses and stores, which ultimately reduced logistics orders across Pakistan and negatively impacted revenue and net profit.

### 4.1.6. Training Issues

People related to the logistics industry in Pakistan are mostly illiterate. On the other hand, organizations also lack fully functional HR departments. This leads to a challenge for logistics companies, for example, how to train drivers and workshop staff for online dispatch and delivery systems. Moreover, the pandemic reshaped socialization, individual collaboration and work and lifestyle [5]. All these issues are challenges for organizations in terms of training their employees for online logistics and supply chain systems in the Pakistani transportation industry. Respondent T1 mentioned this as follows:

> "*Most of them (employees) are illiterate, so it was a big challenge and a hurdle or a bottleneck to provide awareness to these people regarding COVID-19 and change in the work procedures.*"

### 4.1.7. Lack of Relevant Standard Operating Procedures (SOPs)

It was difficult for logistics companies to design SOPs that did not affect the revenues in their business. Invoices, in logistics operations, contain descriptions of goods that need to be checked by different people across the chain, from the product site to the delivery location. This may cause the spread of virus through invoices printed on paper. Furthermore, SOPs are inevitable for drivers' safety who are entering loading premises, i.e., factories, and leaving unloading premises, i.e., warehouses. The creation of appropriate SOPs relevant to the pandemic situation in logistics operations was challenging but inevitable [48].

### 4.1.8. Inefficient Operational Areas

The first phase of COVID-19 initiated a situation of smart lockdown. Laborers and drivers used to travel back and forth to their work and villages/small towns, leading to a shortage of labor on loading and unloading docks, in line with the existing literature [5]. Moreover, production units, especially in the food and beverage industry, were managed in shifts, causing delays in loading finished good stocks in containers and trucks. The labor shortage in warehouses also caused delays in unloading stock. Moreover, a shortage of drivers also affected transit time, which is also evident in the nascent literature [34]. All such issues impacted negatively on the operational areas and caused delays and a decline in performance. Some of the operational issues which affected the transportation sector during the pandemic were highlighted by our respondents (Table 7).

**Table 7.** Operational Issues faced by companies.

| Issues | Description |
|---|---|
| Out of control overhead costs | Every company has overhead costs such as rent, transport, insurance, taxes etc. but it depends on the operations and size of the company. Excessive overhead costs can have a negative impact on a company's profitability. |
| Too much Waste | It is one of the most pressing operational issues in business. When the various resources are used ineffectively or being wasted, it impacts the performance of the organizations. |
| Lack of Performance monitoring | Monitoring the performance of various ongoing operations is very crucial for staff as it helps to resolve any issues and understand the root cause of the problems for possible solutions. |
| Lack of planning | Inadequate planning is a common operational issue particularly during periods of uncertainty. Organizations which fail to pay attention to downward trends, economic conditions and force majeures can land themselves in precarious positions. |
| Unstable Cash flow | Having enough financial resources is critical to manage the supply chain operations. Lack of funds can cause problems such as payment of staff salaries, or buying materials etc. |

### 4.1.9. Performance Management of Drivers

In Pakistani culture, two types of business model can be seen: one is corporate culture business organization and the second one is "Seith" culture business (a Seith is an illiterate person and entrepreneur who runs a transport business). In corporate culture, we can see each department with clearly defined job descriptions for each employee. On the other hand, in "Seith" culture, organizations are headed by one (illiterate) person with no defined job descriptions for employees. In Pakistan, the majority of logistics companies are operating under "Seith" culture, where no concept of monitoring and rewarding for the performance management of drivers and other staff is evident. COVID-19 raised challenges for human resource professionals and "Seiths" to make proper HR policies for drivers to retain, monitor and reward their performance, even in the presence of a high risk of life threats [34].

> "*It was also a big challenge during COVID how to monitor the drivers, how to monitor their performance and incentivize them. So, if you see that we found two types of culture in logistics, one is the corporate culture, the other one is "Seith" (illiterate boss of a company) culture.*"

### 4.1.10. New Entrants in Market–Business Reshaping

In COVID-19, businesses shut down, restructured and/or entered the consumer market. It resulted in a shrunk, squeezed and stretched market share of new entrants. Some of the new entrants in freight logistics in Pakistan are given in Table 8.

**Table 8.** New entrants in freight logistics in Pakistan.

| New Entrants in Freight Logistics | Type |
|---|---|
| Momentum Logistics | Finished goods Containers |
| Pyramid Logistics | Finished goods Containers |
| Faisal Mover Logistics | Open body and Finished goods Containers |
| Keep Trucking Inn | Digital Trucking |
| Truck Sher | Digital Trucking |
| Wahyd Logistics | Digital Trucking |

The borders of all countries were closed to avoid spreading the virus from other countries, creating a situation for many vendors to stop dispatching export products and to think of restructuring of their business models. It was found during this research that a company named "Alpha", working in the vicinity of Faisalabad, which was an exporter of socks to international brands, such as Adidas and Nike, started operations in February 2020 in the logistics industry. Furthermore, there were many other logistics players running operations from passenger buses, and, due to the hold on operations by the Government of Pakistan, switched their core business to freight logistics, such as Prime movers and Faisal mover logistics, which worked only via passenger buses previously. Respondents also mentioned that companies that transported oil switched to finished goods containers. Entering the freight logistics market or switching to it from any other type of business was a major threat and a big challenge for vendors currently operating in the freight logistics market, which ultimately affected their revenues and resulted in a decline in net profits.

> *"We have seen new entrants in the market as their actual businesses went unprofitable or shut down because of COVID-19 restrictions for example in logistic market a new company entered called momentum logistics. The owners of this company had a different business before which was apparel manufacturing. So, such new entrants reduced the market share and profits for the existing companies in this sector."*

### 4.2. Sustainable Strategies

### 4.2.1. Focus on Increased Revenue Post-Pandemic

With the pandemic finally over, it is now time to formulate policies for transportation businesses for increased revenue. The respondents advised some strategies for logistics companies as follows:

- Electronic ads or screens on vehicle containers may help to increase the revenue and revive business post-pandemic.
- Allowing local businesses and brands to paint their brand image and advertisements on containers may also help increase their revenue.
- Transporters can charge per day for using vehicle containers as warehouses in case no space is available in the warehouse.
- Door-to-door on-demand services (last-mile service) could be offered.
- Charging per kilometer, from freight to shipper, could increase revenues.

Respondents R1 and K1 suggested the above recommendations as follows:

> *"So, here is a solution for that. If transporters can get leads for electronic ads or screens on vehicle container so they can add or increase revenues."*

"*Another solution is vehicle containers as warehouse. Basically, it's a new concept. Vehicle containers can be used where is no space in the warehouse by providing transporters per day halting expense. In this way containers are treated as warehouses.*"

Different variants of COVID-19 continued to evolve after the pandemic, and we are not sure if there will be a COVID-19-free environment again. Such outbreaks may occur in the future. Therefore, respondents had a consensus that the only way to increase export and import orders in such situations is that government should make appropriate transportation and logistics policies in general and especially for necessity items, such as edibles, medicines, fruits, cereals, etc., which can be allowed with permits in the case of lockdowns to ensure continued revenue. This will help logistics businesses to run their operations in a smooth way without fear of shutdowns and losses.

### 4.2.2. Contactless Deliveries

Contactless delivery is well recognized around the world [25,49] and, specifically, was boosted in response to the COVID-19 pandemic [50]; however, this is a relatively new phenomenon in Pakistan as logistics worked in a traditional way here and lacked technological advancements. This concept can be further propagated in Pakistan involving goods picked up in a specific contactless area and delivery to the designated location by agents who then inform the customers with a picture of the product at the location. Further benefits of contactless deliveries are the visualization and traceability of health information, short delivery time, small variable delivery cost, smart parcel lockers, unmanned vehicles and fewer human resources required [25]. Governments, organizations and authorities worldwide are focusing on increasing the use of contactless delivery systems, such as unmanned aerial vehicles or drones [50], and this will revolutionize logistics businesses, which will also improve the logistics operations in a developing-economies context.

Respondent I2 stated the following:

"*Less human resources required in contactless deliveries. Smart parcel lockers and unmanned vehicles are fixed investment costs, but it greatly enhances visualization, traceability and also protects the health of crew.*"

### 4.2.3. Expansion of E-Commerce Business Is Not a Threat to Transportation Business

Respondents opined that during COVID-19, increasing online stores due to lockdown in Pakistan and across the globe were no threat to transportation businesses. They only changed the mode of transportation. In other words, primary logistics business is transferred to secondary logistics business. Hence, there will be a decrease in revenues in primary logistics and an increase in revenues in secondary logistics. It also impacts local market consumer orders but leads to an increase in online store orders, i.e., food panda, daraz, airlift, etc. Respondent L3 described this as follows:

"*It is not a threat to logistic business. Actually, it is a transfer of business from primary logistics to secondary logistics.*"

Respondents revealed that tech giants are making digital apps through collaboration with big transportation companies and finding new ways of trucking with the help of block chain to gain more and more controls over shipments in Pakistan. This creates a need for more educated people to work in such environments, who will ultimately replace illiterate people in the supply chain industry [40]. It will lead to better human resource department engagement, development and training people and the culture in logistics companies. Considerable research is evident in the areas of online shopping and e-commerce [6], which is a much-needed change in the logistics sector in Pakistan.

### 4.2.4. Tech-Based Performance Management of Drivers

Some types of jobs are not very rewarding in the context of developing countries, such as security guards or helpers, etc. Heavy truck drivers are no different. The respondents revealed that drivers are paid less, even though they travel thousands of miles and stay

on roads for weeks to deliver stock. This may impact their well-being and, thus, their performance. Their performance is also not monitored or evaluated properly by companies. Few transportation companies have proper human resources departments and implement policies for drivers' performance management [5]. Large firms, institutes and transportation companies have taken the initiative to offer training to drivers as well as also starting to monitor drivers' performances. The Motorway Police have also established the National Highways & Motorway Police Training College, Sheikhupura, to train heavy vehicle drivers. A few big transport companies, such as Capital Marketing Services (CMS), Bashir Sadiq Logistics (BSL), Daewoo express, etc., had developed driver training schools, for short courses related to driving manuals for different vehicle, health and safety standards related to on-road driving and on-road sense of driving, etc.

Some transport companies track their fleets in collaboration with tracker companies, such as E-Drive, Nexer Tech, TPL, etc., and they have already given online web-based and mobile portals to customers for multiple types of updates and notifications, which ultimately helps FMCGs and transporters to verify and gauge the performance management of drivers during driving and idling throughout their journey. However, still, there is a lack of proper performance management for drivers in the logistics sector in Pakistan, and there needs to be an industry-wide implementation across all sizes of company. The respondents mentioned some key parameters through which companies can check and incentivize the performance management of drivers, as given in Table 9. The respondents further advised that data for the performance management of drivers can be checked and fetched through the Internet of Things (IOT) and tracker portals. IOT can play an important role in increasing the efficiency of the supply chain in Pakistan if appropriate strategies are developed [17].

**Table 9.** Performance management indicators.

| | |
|---|---|
| 1. | Kilometer travelled on daily and monthly basis. |
| 2. | Harsh braking during driving hour. |
| 3. | No of times over speeding during a month. |
| 4. | Traffic police and excise and taxation Challans. |
| 5. | Idling time during journey or a complete round trip. |
| 6. | Achieving transit time of a complete trip. |
| 7. | No of trips during a month verses target. |
| 8. | Revenue generated by each driver during a month. |
| 9. | Halting time throughout month. |
| 10. | Workshop expenses throughout month. |
| 11. | Diesel consumption and analysis. |
| 12. | No of trips in hilly areas throughout the month. |

Large transport companies maintain their operations room to report big fleet size vehicles, where they use tracker data to maintain different types of reports, such as vehicle movement reports and trip-closing reports. The shortage of drivers could be managed properly [34] in this way. It is also evident that technological adoption in the supply chain and logistics industry will have a positive impact on sustainability [15]. Based on the responses received, we suggest relay logistics, in which drivers carry the loaded truck for some kilometers without taking a rest. After covering a specific milage, the driver is replaced with another driver who will drive the truck for the rest of the journey to the destination. For instance, considering the 1600 KM long route from Karachi to Islamabad, the relay process is demonstrated in Table 10. In this way, the transit times can be managed efficiently. Moreover, the shortage of labor on loading and unloading docks creates high loading and unloading times. The strategy advised in this case is to change the manual

loading process with palletized loading with the help of lifters in warehouses and factories, which will reduce loading and unloading times as well as reductions in wages.

**Table 10.** Concept of relay logistics for one complete round trip.

| (Karachi to Islamabad) | |
|---|---|
| No of Kms Travel | Driver Changed as per defined Kms |
| 0---------------400 Kms | Driver 1 |
| 401------------800 Kms | Driver 2 |
| 801------------1200 Kms | Driver 3 |
| 1201----------1600 Kms | Driver 4 |
| No of Kms Travel | Driver Changed as per defined Kms |
| 0---------------400 Kms | Driver 4 |
| 401------------800 Kms | Driver 3 |
| 801------------1200 Kms | Driver 2 |
| 1201----------1600 Kms | Driver 1 |

### 4.2.5. Digital Trucking

Post-pandemic, digital trucking seems to be the new game changer in Pakistan's logistics industry. In recent times, due to increased competition in the supply chain industry, high customer demands and keeping records of each transit, multiple companies entered the model of digital trucking in Pakistan. Respondents revealed some of the features such as electronic proof of delivery (ePOD) (Table 11), smart contracts, record maintenance, easy payments etc. which made digital trucking more successful in Pakistan during COVID-19 and which ultimately creates a huge challenge for old logistics players to survive post-pandemic. Respondent S1 mentioned this as follows:

> *"If we see, during the pandemic, digital tracking concept emerged. So, in conventional concept it is proof of delivery (POD) which is a document signed and stamped manually by a factory manager. When shipment reaches the warehouse, it is again manually signed and stamped by the warehouse. When the stock is dispatched from there, the transporter signs and stamps on that proof of delivery and so on. . .., the pandemic brough in technology (electronic proof of delivery) and concept of contactless deliveries which is very new here in Pakistan and is good initiative even after pandemic."*

From this perspective, participants mentioned that it is important to note that transporters who have big fleets of trucks and trailers maintain their vehicles under their own workshops, and they have direct contracts with shippers, such as FMCGs, oil companies, manufacturing firms, etc. Thus, these transporters have their own tracking portals to trace shipments and stay updated about vehicle journeys through their control rooms. So, digital trucking has no impact on big fleet transporters, but those transporters who have a small fleet size or have a single-vehicle fleet can benefit from digital trucking and obtain better freight offers than on the open market, without wasting their time in the search for loads with the help of local truck stations called "addas". Furthermore, these single-fleet transporters do not have tracking portals to trace shipments and do not remain updated about vehicles. Through digital trucking, they can install portable tracker devices and keep customers updated about the status of their shipment. Based on the above discussion, we developed the following framework (Figure 1), which shows how tech-based strategies can resolve the challenges for logistics companies in a developing-economy context.

**Table 11.** Digital trucking features.

| Digital Trucking Features | Beneficiaries |
| --- | --- |
| Electronic proof of delivery (ePOD) | Gains for Shipper. |
| Improved cash flows for clients. | Gains for Transporter. |
| Digital Agreement | Mutual gains for shipper and Transporter. |
| Smart Contract | Mutual gains for shipper and Transporter. |
| Record of freight history | Mutual gains for shipper and Transporter |
| Open Enterprise Logistic Models (OEL Model) | Mutual gains for shipper and Transporter |
| Subcontracting 3PL contracts | Gains for Transporter |
| Block chain | Mutual gains for broker, shipper and Transporter. |
| Guarantee business availability | Gains for Transporter. |
| Fleet Management solutions | Gains for Transporter. |
| Affordable prices point | Gains for Shipper. |
| Simplified entire value chain | Mutual gains for broker, shipper and Transporter. |
| Transparent costing | Mutual gains for shipper and Transporter. |
| Dashboard for all your previous and upcoming bookings | Mutual gains for shipper and Transporter. |
| Secure and convenient payment methods | Mutual gains for shipper and Transporter. |
| Updates and notifications | Mutual gains for shipper and Transporter. |
| Cost saving | Gains for Shipper. |
| Flexibility and reliability | Gains for Shipper. |
| Live tracking and digital insights | Gains for Shipper. |
| Reduced carbon footprints | Gains for Shipper. |
| Revenue boost | Gains for Transporter. |
| Easy Payments | Gains for Transporter. |
| Reduced payment times | Gains for Transporter. |
| Value-added services | Gains for Transporter. |

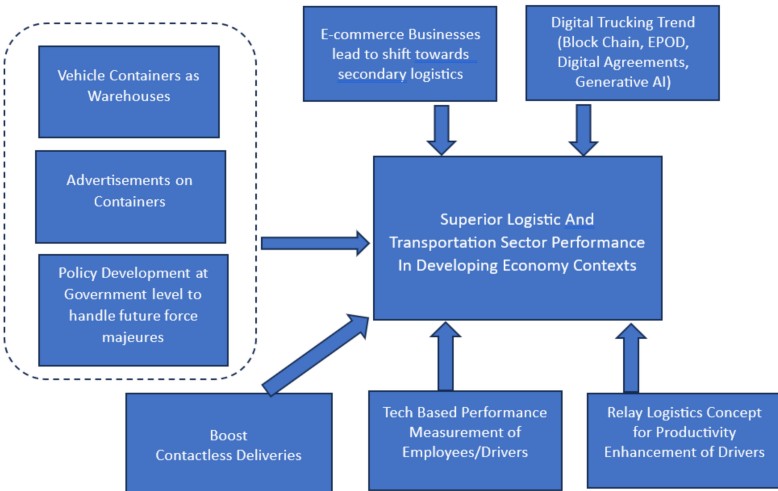

**Figure 1.** Framework to address the logistics challenges using tech strategies.

## 5. Conclusions

Realizing logistics as being a key pillar in economic growth and development [2,51], this study identified the challenges faced by logistics companies during and post-COVID-19 and some relevant strategies to handle these challenges post-COVID-19. Based on this, we found that logistics and the supply chain industry faced several challenges during the pandemic, such as a decline in revenue, reduced import, export and local businesses, limited supply, increase in e-commerce business, new entrants in the market, lack of SOPs, inefficient operational areas, performance management and training of drivers. Various sustainable strategies have been identified to overcome these challenges, in the light of analysis performed on respondents' views. These strategies include contactless deliveries, expansion in e-commerce business, tech-based performance management of drivers, digital trucking and increasing revenue through utilizing containers, such as warehouses, digital marketing, etc.

### 5.1. Theoretical and Managerial Implications

Theoretically, this study contributes to the existing body of knowledge on developing country perspectives on the challenges faced by the transportation industry generally and the supply chain specifically and adds to the scarce research in this study area in the unique COVID-19 context. This research work adds to the supply chain literature by proposing context-relevant strategies that could ensure sustainability in using available resources in order to ensure reliability in the supply chain industry. These integrated research dimensions complement this study's contribution with the intention to achieve resilient, sustainable and performant supply chains in a developing-economy context. Managers and practitioners in logistics and the supply chain industry in Pakistan need to reconnect the dots and refine their existing strategies in light of the findings of this study. In this vein, contactless deliveries are the most efficient and appropriate concept to demonstrate todays' requirements by the industry and customers [40] using means, such as unmanned vehicles and automation systems [50]. Digital trucking is a way forward for smart and efficient logistics and the supply chain. Tech-based performance management and training of drivers will make them aware and motivate them for their efficient work in collaboration with the latest technology. Practitioners are required to consider expansion in e-commerce as a new business dimension instead of a threat, along with a focus on appropriate changes in logistics and the safety of workers [6]. The findings of this study will also directly and indirectly serve the CPEC, the most important project for Pakistan [15,16], as well as for many other countries including China. The CPEC will not only bring better road infrastructure but will be a game changer in terms of new business opportunities for logistics companies. Thus, efficient implementation of these strategies will help these companies gain success in an effective way. Thus, the findings of this study facilitate all stakeholders, i.e., local industry, government and CPEC authorities, to shape their strategies, especially after COVID-19, when businesses are reviving.

### 5.2. Limitations and Future Research Directions

This research work is not without limitations, like any other study. It is based on freight logistics, inbound logistics and outbound logistics and, thus, cannot be generalized on passenger bus logistics. The COVID-19 pandemic was first experienced by everyone, so it was difficult to make immediate policies and regulations to handle the situation. In transport companies, most decisions are made real time according to situations, such as accidents, loss of life in COVID-19, strikes and roadblock demands; therefore, a separate study to demonstrate strategies for such situations is imperative. In addition, the findings of this study are very pertinent to the logistics and supply chain of Pakistan and, hence, cannot be generalized to other contexts. Future research should focus on expanded research designs to cover all aspects of the transportation sector. Despite all the benefits [50], the findings of this study highlight the neglected area of contactless deliveries in Pakistan. Future research with a focus on contactless deliveries' effectiveness in developing countries

has great potential to foster the role of emerging technologies. Similarly, future research can be conducted on the tech-based performance management of personnel working in the transportation industry and digital trucking. This study focuses on the sustainable supply chain and logistics and, thus, the emerging concept of a green supply chain [1,18,52,53], and logistics can be linked with these emerging strategies identified in this study.

**Author Contributions:** Conceptualization, M.S.S.; Methodology, M.S.S.; Formal analysis, W.A.; Investigation, H.S.; Resources, H.S.; Writing—original draft, W.A.; Writing—review & editing, F.C.; Supervision, F.C. All authors have read and agreed to the published version of the manuscript.

**Funding:** This research was funded by Macau University of Science and Technology Faculty Research Grants (FRG), grant number FRG-22-108-MSB, and The Macau Foundation Project (MFP), grant number MF-23-008-R.

**Data Availability Statement:** Not applicable.

**Acknowledgments:** The authors would like to extend their sincere appreciation to the support by grants from Macau University of Science and Technology Faculty Research Grants (FRG) under grant number FRG-22-108-MSB and The Macau Foundation Project (MFP) under grant number MF-23-008-R.

**Conflicts of Interest:** The authors declare no conflict of interest.

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
