# Peer review of "Sustainable Technology Strategies for Transportation and Logistics Challenges: An Implementation Feasibility Study"

_sustainability, doi:10.3390/su152115224_

Round 1
Reviewer 1 Report
- An outline of the paper should be added at the end of the introduction.
- The article's contribution to the literature should be stated more clearly in the introduction section.
- More clear and detailed information should be given about why the methods or approaches preferred in the study were chosen.
- The authors claim to manage the challenges of the transportation and supply chain industry. However, more descriptive information about these challenges should be given in the introduction section.
- Although sufficient literature research is provided in the study, more detailed and clear information should be given about the gap that the study fills in the literature.
- Methodology, Row 153: The paragraph should be continued appropriately.
- An explanation of concepts such as H1, L1, R2, and T1 included in the study should be given.
Reviewer 2 Report
1. I would suggest the change in the title: A sustainable strategy for logistic transportation challenged by the Covid-19 and post-Covid-19: an implementation feasibility analysis.
2. The authors should also present in a table a point-to-point comparison between logistic transportation challenges, sustainable strategies, and its implementation feasibility analysis.
3. The implementation feasibility analysis should also be explained in a paragraph too in Section 4 and 5.
-
Reviewer 3 Report
The paper is based on empirical research based on interviews with different classes of stakeholders belonging to the logistics business. The interviews cover aspects related to strengths and weaknesses, opportunities and threats of the industry, in the era of COVID-19. The paper is well written and fluent, however, I suggest minor changes:
1) the sample is small, 20 interviews. How is the appropriateness of the small number of interviews justified? Are there previous studies in the literature that support this? It is necessary to justify this choice;
2) fix the style of the tables (the font size is not uniform) and their location (tables often interrupt the text of the sections, they should be placed in different places);
3) final proofreading and language revision is recommended.
No relevant comments.
Round 2
Reviewer 1 Report
I think that this form of the manuscript is suitable for publishing in your journal. I am grateful to the authors for their efforts.